# Rapid and Accurate Approach for Honeybee Pollen Analysis Using ED-XRF and FTIR Spectroscopy

**DOI:** 10.3390/molecules26196024

**Published:** 2021-10-04

**Authors:** Agata Swiatly-Blaszkiewicz, Dagmara Pietkiewicz, Jan Matysiak, Barbara Czech-Szczapa, Katarzyna Cichocka, Bogumiła Kupcewicz

**Affiliations:** 1Department of Inorganic and Analytical Chemistry, Faculty of Pharmacy, Collegium Medicum in Bydgoszcz, Nicolaus Copernicus University in Toruń, Jurasza 2, 85-089 Bydgoszcz, Poland; kupcewicz@cm.umk.pl; 2Department of Inorganic and Analytical Chemistry, Poznan University of Medical Sciences, Grunwaldzka 6, 60-780 Poznan, Poland; dagmarapietkiewicz3@gmail.com (D.P.); jmatysiak@ump.edu.pl (J.M.); 3Epidemiology Unit, Department of Preventive Medicine, Poznan University of Medical Sciences, Święcickiego 6, 60-781 Poznan, Poland; bczechszczapa@ump.edu.pl; 4Department of Pediatric Gastroenterology and Metabolic Diseases, Institute of Pediatrics, Poznan University of Medical Sciences, Szpitalna 27/33, 60-572 Poznan, Poland; kcichocka@ump.edu.pl

**Keywords:** bee pollen, spectroscopy, 2T2D correlation spectroscopy, multi-elemental analysis

## Abstract

Since honeybee pollen is considered a “perfectly complete food” and is characterized by many beneficial properties (anti-inflammatory, antioxidant, anti-bacterial, etc.), it has begun to be used for therapeutic purposes. Consequently, there is a high need to develop methods for controlling its composition. A thorough bee pollen analysis can be very informative regarding its safety for consumption, the variability of its composition, its biogeographical origin, or harvest date. Therefore, in this study, two reliable and non-destructive spectroscopy methods, i.e., ED-XRF and ATR–FTIR, are proposed as a fast approach to characterize bee pollen. The collected samples were derived from apiaries located in west-central Poland. Additionally, some commercially available samples were analyzed. The applied methodology was optimized and combined with sophisticated chemometric tools. Data derived from IR analyses were also subjected to two-dimensional correlation spectroscopy. The developed ED-XRF method allowed the reliable quantification of eight macro- and micro-nutrients, while organic components were characterized by IR spectroscopy. Principal component analysis, cluster analysis, and obtained synchronous and asynchronous maps allowed the study of component changes occurring dependently on the date and location of harvest. The proposed approach proved to be an excellent tool to monitor the variability of the inorganic and organic content of bee pollen.

## 1. Introduction

Thorough analyses of honeybee products (including honey, pollen, propolis, etc.) are supposed to provide essential information useful in multiple fields, such as food nutrition, environment pollution, toxicology. Multi-elemental analysis of macronutrients (i.e., K, P, S) and micronutrients (i.e., Fe, Zn) may provide knowledge about their nutritional value. In contrast, the examination of heavy metals like Cu is usually performed to determine products’ toxicity and contamination. The characterization of the spectroscopic properties of bee pollen may extend our knowledge about apiculture development and differentiate species of flowers explored by bees. Moreover, the combination of data from the analysis of both inorganic and organic components can be helpful in the biogeographical origin determination of the analyzed products [1,2]. 

Honeybee pollen is a source of protein, lipids, and other nutrients like minerals and vitamins. It is composed of different flower pollen shaped into granules by bee enzymes from salivary glands, honey, and nectar [3]. Due to its complex composition, bee pollen is considered a “perfectly complete food” [4]. Therefore, this product is of growing interest to people who care about their health. Natural products like pollen are becoming increasingly popular as alternative drugs for their dietetic and therapeutic qualities. Bee pollen is recommended as a supplement for active people and patients during recovery. Many papers described a wide range of pollen beneficial features, including anti-inflammatory [5], anti-atherogenic [3], antioxidant [6], anti-bacterial [7], hepatoprotective [8] properties. Since pollen quality affects bees and humans, there is a high need to develop and standardize effective methods for analyzing and controlling its composition.

Considering how useful, informative, and constructive it may be to analyze bee pollen, the application of a proper approach for this purpose is essential. There are several analytical methods for inorganic elemental analysis. The most common and universal methods are inductively coupled plasma mass spectrometry (ICP-MS) and inductively coupled plasma optical emission spectrometry (ICP-OES). Biological samples are usually digested and heated to high temperatures to obtain the most accurate results. Otherwise, some solid matter like tissues or components like lipids, proteins, or cells, might stick to the spectrometer’s surface and cause problems with ionization [9]. Since ICP-MS and ICP-OES require mineralization of the samples, these methods are time-consuming and tedious and generate a significant waste stream [10]. Therefore, we decided to use energy-dispersive X-ray Fluorescence (ED-XRF) spectroscopy for bee pollen multi-elemental analysis in this study. During ED-XRF analysis, photons of energy are generated by an X-ray tube and collide with an atom. Then, an inner shelter is knocked out, leaving a hole, and an outer shelter electron fills the vacancy in the lower orbital. As a result a fluorescent X-ray (energy) is realeased [11,12]. This study used the ED-XRF technology to analyze the most abundant and essential minerals (K, P, Ca, Zn, Mn, Fe, Cu, and S) in bee pollen, vital for human health and well-being. Recently, our group has published a paper describing the use of ICP-MS and ICP-OES to analyze a large group of bee pollen samples, including propolis, bee pollen, and royal jelly [13]. However, compared with the ICP methods, ED-XRF is much faster and does not require a tedious sample preparation. The use of ED-XRF enables the analysis of essential inorganic components. Therefore, this technique seems to be a powerful screening tool. 

The application of the ED-XRF technology allows for reliable multi-elemental analysis. To obtain a complete picture of the bee pollen components, we used in this study another non-destructive method. Fourier transform infrared (FTIR) spectroscopy with attenuated total reflectance (ATR) is currently available to characterize the structure and chemical composition of bee products samples. A FTIR spectrum is defined as a characteristic “molecular fingerprint” that can be used to identify organic components, including chemical bonds. Several papers reported the possibility of analyzing bee products using FTIR. Most of them focused on the characterization of products of botanical origin [14,15]. FTIR data are often combined with morphological and physicochemical properties of bee products [2]. The use of advanced chemometric tools also enables the authentication of the analyzed samples and control the quality of the products [14,16]. 

In the literature, other analytical methods can be applied to characterize bee pollen’s composition [16,17,18,19,20]. One of the most frequently used techniques is high-performance liquid chromatography (HPLC). It was proved to be an effective and reliable method to determine phenolic compounds [16] and amino acids [20] in bee pollen samples. Therefore, HPLC can be used as an authentication technique for this product. However, in this study, the main concerns were a short analysis time and the non-destructive character of the analytical method. Since FTIR enables screening the whole molecular profile of a sample, it seemed to be a proper strategy for this research. Moreover, bearing in mind that macro-and micro-nutrients are essential for consumers, the quality of the bee pollen should also be examined regarding inorganic components. Therefore, ED-XRF and FTIR spectroscopy can be applied as complementary methods to monitor bee pollen variability.

This study aimed to propose a quick and reliable approach to characterize bee pollen composition without destroying the sample. For this purpose, we analyzed bee pollen samples collected in Poland (Greater Poland region) in 2018 and 2019 and a group of commercially available products. Two samples, first examined using ICP-MS and ICP-OES [13], were used as standard samples to develop a proper ED-XRF method. Two different ED-XRF approaches were tested. The comparison between the results provided by ED-XRF and those from ICP-MS/ICP-OES enabled the evaluation of the usefulness of the presented methodology. Then, we analyzed the remaining samples using the most suitable ED-XRF approach. To the best of our knowledge, the ED-XRF technique has never been applied to bee pollen analysis before. ATR–FTIR spectroscopy coupled with two-trace two-dimensional (2T2D) spectroscopy was used to characterize organic pollen components comprehensively. A fast and practical analytical strategy is proposed, combining ED-XRF, FTIR, and proper sophisticated chemometric procedures permitting bee pollen samples characterization. 

## 2. Results 

### 2.1. Bee Pollen Multi-Elemental Analysis

#### 2.1.1. ED-XRF Quantitative Analysis of the Samples P1–P5 Using a Standard Sample

ICP-MS and ICP-OES were used to analyze and quantify elements in two bee pollen samples (P2 and P5). Therefore, in the next step, sample P5 was used as the standard sample to develop the ED-XRF quantitative method for bee pollen analysis. Firstly, sensitivity coefficients were calculated using a standard sample, and then the unknown samples were analyzed using those sensitivity coefficients. The samples P2 and P5 were also further used to evaluate the reliability of the ED-XRF method. The comparison of the results obtained using different methods is presented in Table 1. 

The newly applied ED-XRF quantitative method was used to quantify elements in the five pollen samples (P1–P5) in two forms: whole pollen granules and powdered pollen. Firstly, we analyzed the whole pollen granules. The percentage errors between the results obtained with ICP-MS/ICP-OES and ED-XRF for samples P2 and P5 were between 0.3% and 19.9%, except for Fe, whose error reached 30.4% for sample P2 (Appendix A). The relative standard deviation (RSD) calculated for three ED-XRF measurements was 0.8–15.1% for all five samples. Only RSD calculated for the measurement of Mn in sample P2 was higher, corresponding to 22.2%. 

The results from the ED-XRF analysis of the powdered bee pollen samples were comparable to those of whole granules analysis (Appendix A). However, the percentage errors concerning the ICP-MS/ICP-OES analysis were higher and ranged from 3.3% to 29.6%. The exception was Cu concentrations, which were much higher in the ED-XRF analysis. Therefore, the errors were 131.6% and 143.9%. Unexpectedly, there was also a large difference in Ca concentration in P5, which was 41.9% in this sample. RSD calculated for three ED-XRF measurements was 0.3–8.4%. 

#### 2.1.2. ED-XRF Qual-Quantitative Analysis of the Samples (P1–P5) without a Standard Sample

A qual–quantitative method for bee pollen analysis was designed based on ED-XRF. A qual-quantitative analysis enables the identification and quantification of the elements contained in a sample. However, without a standard sample or calibration curve, the results could be inaccurate. In addition to the previously presented results, the qual-quantitative method allowed the identification of chlorine in all samples and of silicone in sample P1. The obtained results were further compared to the results from ICP-MS/ICP-OES analysis and ED-XRF quantitative analysis. 

The results of whole pollen granules analyses obtained with the two ED-XRF methods differed (Table 1, Appendix A). The percentage errors range was 0.1–35.1%. Higher levels were only reached for Cu (72.9–108.8%) and P (26.6–40.3%). The percentage errors for the results obtained with ICP-MS/ICP-OES and ED-XRF for samples P2 and P5 were mainly between 0.8% and 32.9%, except for Cu (between 41.5% and 96.2%). The RSD values for three measurements were in the range 0.4–25.6%.

The concentrations for powdered pollen determined by this method differed from those obtained with quantitative analysis by a value from 0.1% to 18.4% (Appendix A). The exception was the percentage error for S, which reached the values 29.6–54.8%. Percentage errors for the results obtained from ICP-MS/ICP-OES and ED-XRF for the samples P2 and P5 were mainly between 0.3% and 26.3%. The exception was Cu, for which higher errors were reported. The RSD values the for ED-XRF measures were from 0.5% to 14.3%.

#### 2.1.3. ED-XRF Quantitative Analysis (Using a Standard Sample) of Commercially Available Pollen Samples

We selected a quantitative method with a standard sample as the most reliable technique for analyzing bee pollen. The results also assumed that the whole pollen granules were more suitable for this experiment than powder pollen. Therefore, the concentrations of K, P, Ca, Zn, Mn, Fe, Cu, and S were measured in whole pollen grains using the previously developed method. The results are presented in Appendix A. The RSD for three measures was below 10.0%, with only a few exceptions.

Interestingly, two manufacturers declared some estimated concentrations of the studied minerals. However, they noticed that these values might differ depending on the batch of the product. Moreover, the method used to quantify these elements was not indicated. For the sample K3, the concentrations obtained with the ED-XRF technique and those reported on the product label varied between 2.7 and 29.9%. For the sample K5, the differences were much more significant (16.6–79.0%). The comparison between all analyzed samples is presented in Figure 1.

#### 2.1.4. Principal Component Analysis and Cluster Analysis

A hierarchical cluster analysis was performed to identify similar groups among the pollen samples. Four clusters were grouped based on the ED-XRF data, as presented in Appendix A. Interestingly, the sample harvested in 2018 (P1) constituted a separated cluster. The samples collected in the same beehive were grouped separately, on the basis of the harvest date. Principal component analysis (PCA) confirmed the clustering results. A biplot graph including samples and analyzed elements is presented in Figure 2. Detailed information about PCA analysis performed on the data from ED-XRF analysis is shown in Appendix A. 

### 2.2. ATR-FTIR Spectroscopy

Mainly, bee pollen is composed of proteins, saccharides, and lipids; therefore, IR spectra (Figure 3) were characterized by very similar peaks. Due to minor differences in the chemical composition of each pollen, several peculiarities could be observed. The prominent IR peaks with their position (wavenumber), vibrational mode, and biochemical assignment are presented in Table 2. Detailed information and description of the IR bee pollen spectra have already been reported [2,14]. Since the peaks zone of 3500–1800 cm^−1^ was usually similar for all samples, the spectral region 1800–650 cm^−1^ was used for chemometric analysis. The peaks at positions 3500–1800 cm^−1^ were assigned to cellulose, lipids, water, and proteins.

#### ATR–FTIR Spectroscopy Chemometric Analysis

Firstly, hierarchical cluster analysis and PCA were performed (Figure 4 and Appendix A). Three main clusters were generated. It should be emphasized that samples K1 and K2 (from the same producer) were in the same cluster. Similarly, samples P1–P3 and P5 (from the same beehive) were in the same group. However, one sample, i.e., P4, differed significantly from the others. This can also be observed in the PCA diagram, where the distance between P4 and the other samples is noticeable. Detailed information about the PCA analysis performed on the ATR–FTIR spectra is presented in Appendix A.

Further analysis was based on the construction of synchronous and asynchronous maps (Figure 5 and Figure 6). The basis and theory of 2T2D correlation spectroscopy were described by Noda [21]. Correlation analysis proceeded only on the samples from the same beehive (P1–P5). This was because of the differences between them resulted only from the date of harvest. The more similar the samples were to each other, the more similar their synchronous maps looked. Samples P1, P2, and P5 mainly differed in auto-peaks around 1652 cm^−1^ (amide I of proteins) and 1735 cm^−1^ (lipids and hemicellulose). It can be observed that, besides the strong auto-peaks around 1652 cm^−1^ and 1735 cm^−1^, sample P3 also had a strong auto-peak around 1515 cm^−1^ (phenolic acids), while the synchronous map for sample P4 showed additional auto-peaks at a position around 1350 cm^−1^ (amide III of proteins) and a cross-peak in the range of 1300–1500 cm^−1^.

Asynchronous maps were generated using sample P1 as a reference. The differences between samples P1 and P2, as well as P1 and P5, were rather small. They mainly occurred at the peak positions around 1000–1200 cm^−1^ (sugar and proteins), 1652 cm^−1^ (amide I of proteins), and 1735 cm^−1^ (lipids and hemicellulose). On the map for sample P3, only small differences in the peak position can be observed around 1000–1200cm^−1^. However, greater changes are visible at the peak positions 1514 cm^−1^ and 1735 cm^−1^. The asynchronous map also confirmed that sample P4 significantly differed from the others. It presented strong peaks in the positions around 1000–1200 cm^−1^, 1200–1300 cm^−1^, 1380–1400 cm^−1^, 1514 cm^−1^, 1735 cm^−1^, and 1800 cm^−1^. Therefore, it can be assumed that sample P4 differed from the other bee pollen samples from the same beehive in terms of sugars, proteins, and lipids composition. Nevertheless, based on the maps from correlation spectroscopy, sample P3 appeared quite similar to sample P4, in agreement with the harvest date.

## 3. Discussion

In this study, we introduced ED-XRF as a proper method for multi-elemental bee pollen analysis. Several advantages characterize this method. One of the essential features that were considered is the non-destructive nature of the analysis. Since there is a significant need to perform “greener” laboratory practices, it is crucial to minimize sample preparation and, hence, chemical and consumable waste generation [12]. Additionally, many papers described ED-XRF as an accurate and rapid technique. ED-XRF has been already widely used for the verification of the geographical origin of olive oils [22] and different kinds of honey [23], the authentication and identification of drugs impurities [24,25], the elemental analysis of food products [12,26]. An analysis of the literature allowed us to find several drawbacks in previously conducted studies. First of all, many studies did not confirm their results with different quantification approaches.

Moreover, there is a very poor commercial availability of the reference materials and standard samples used. Therefore, ED-XRF spectrometers are often treated as “black boxes”. Reference materials are not used or they are not matched to the studied samples and their matrices. It should be emphasized that matrix effects could have a considerable impact on the results. Regarding biological samples, the applied calibration standards should be optimized for carbon matrices [12]. To avoid these inconsistencies, we decided to use comparative methods in our study: ICP-MS and ICP-OES on one hand and different ED-XRD approaches on the other, i.e., a quantitative one with a standard sample and a qualitative–quantitative one without a standard sample.

A proper comparison of the quantitative results from bee pollen analysis seems to be quite a challenging task. Honeybees collect different amounts of pollen from selected species of flowering plants [27]. Therefore, the obtained pollen is highly heterogeneous and complex. Thus, quantitative data derived even from a very accurate technique like ICP-MS may be burdened with error. Despite these inconveniences, our ICP-MS/ICP-OES and ED-XRF results were quite similar, and the percentage errors were relatively low. Surprisingly, the analysis of whole pollen granules was characterized by better accuracy than the analysis of the powdered samples. Most of the papers described the analysis of pressed pellets or samples crushed in a mortar [12,24,25]. The possibility of obtaining accurate results when analyzing whole pollen granules allowed us to use ED-XRF to monitor the elemental composition of pollen from different hives or different producers without any influence on the sample. An interesting approach for quantitative ED-XRF analysis is also the use of internal calibrators instead of standard samples. However, a publication by L. Herreros-Chavez et al. [26] showed that the results provided by this method did not resemble those obtained by ICP-OES.

As expected, the qual–quantitative method was characterized by lower accuracy than the quantitative method. First of all, the qual–quantitative approach was conducted without any standard sample or calibrators. Therefore, the quantitative analysis was conducted based on the theoretically calculated fluorescent X-ray intensities. The obtained results may be treated as semi-quantitative [22] and can be used to compare pollen of different origins or collected in different seasons. In addition, the great advantage of the qual–quantitative method is the identification of the unknown elemental composition.

Bee pollen is not only a nutritional food for bees but also a natural dietary supplement for humans. Due to its broad-ranging health benefits, it is classified as a “superfood.” Bee pollen can be found in different forms on the market, e.g., in powder, pellets, granules, and capsules [28]. Therefore, in this study, also commercially available pollens were analyzed. The obtained results were different for the elements compared to the results for the non-commercial, harvested samples, most likely due to the products’ origin and time of harvest. However, the measured concentrations for all samples in this study are in line with those reported by previous papers [4,29,30]. K and P were the most abundant, while minerals with the lowest concentrations were Mn and Cu. The high concentrations of K and P are justified because these minerals are necessary for the proper development of insects [31]. Interestingly, in the dendrogram and PCA, it is noticeable that the composition of the sample harvested in 2018 significantly differed from that of the other samples.

ED-XRF is a novel approach for bee pollen analysis. However, to obtain more detailed results, the complementary spectroscopy method ATR–FTIR was proposed in this paper. The great advantage of this method is its simplicity. Moreover, it does not require a specific, long sample preparation. It is fast and cost-effective. Several papers reported the use of IR to characterize and identify pollens (including bee pollen) and bee products. It has proven to be a reliable method for the identification of plants as well as environmental conditions [32]. IR was also successfully used as a rapid tool to characterize aeroallergens in pollens [33]. Many publications confirmed that it is a valuable method to characterize pollen’s structure and chemical composition [34,35,36]. According to bee products, IR analysis of pollen and honey was used for the authentication of unifloral rape honey [14]. Recently, a novel approach for bee pollen analysis combing HPLC and IR was described [16]. IR-based metabolic profiles were compared with results derived from the HPLC technique and subjected to multivariate analysis. Anjos et al. emphasized that there is a need to label bee pollen products correctly for customer’s convenience and health. Therefore, these authors proposed the IR methodology to analyze essential components for the nutritional labeling of this product [34]. The most meaningful FTIR peaks presented in the mentioned papers are in line with our results. However, to better characterize the variability in pollen composition, a sophisticated 2T2D approach was applied.

Conventional FTIR spectroscopy generates 1D linear spectra. However, in seemingly very similar profiles, especially of complex samples, it is very tedious to perform a detailed analysis. Therefore, the application of 2D spectroscopy allows us to identify outlier development. Recently, it has become a powerful tool for the interpretation and visualization of structural changes in complex matrices [21]. The application of 2D IR spectroscopy has already been successful for the identification of different kinds of pollen (cattail pollen, pine pollen, and bee pollen). This innovative approach allowed the visualization of more differences in the IR spectra than conventional analysis [37].

This study proposed a 2T2D approach to monitor the most remarkable changes in bee pollen samples derived from one beehive. Considering that the harvest date is crucial for pollen composition [38], samples P1–P5 were analyzed using asynchronous and synchronous 2T2D correlation spectra. The obtained 2D spectra have a significantly increased resolution compared to 1D spectra and contain important information about samples’ molecular structure [39]. The generated results allowed to underline differences, as well as similarities, between samples. Our findings, which concerned substantial changes in bee pollen composition resulting only from different harvest dates, suggest that commonly available bee products should be constantly monitored.

## 4. Materials and Methods

### 4.1. Samples

For this study, five samples of honeybee pollen were collected from beehives located in Greater Poland Voivodeship in west-central Poland, on the following days: 12 August 2018 (P1); 2 June 2019 (P2); 14 June 2019 (P3); 29 June 2019 (P4), and 20 July 2019 (P5). Samples were stored in the 15 mL tubes at −20 °C until required. Additionally, seven commercially available samples (K1–K7) were purchased from a grocery and analyzed using ED-XRF. Samples K1–K7 were derived mainly from Polish beehives, except for sample K4, which was from Italy. Samples K1 and K2 were from the same manufacturer but were from different production series.

### 4.2. ICP-MS and ICP-OES Analysis

Two samples (P2 and P5) were firstly analyzed using ICP-MS and ICP-OES. Sample preparation consisted in microwave-assisted digestion and mineralization that was conducted in PTFE closed vessels. The mineralization process lasted 20 min at 180 °C. Centrifuge tubes with minerals were filled to the mark with deionized water and then subjected to filtration. The obtained solutions were diluted with deionized water. A quadrupole ICP-MS 7800 (Agilent Technologies, Tokyo, Japan) was used to determine trace elements and heavy metals. ICP-OES analyses were performed to determine the main elements’ content in the analyzed beehive products. The method was validated for linearity, limits of detection and quantification, precision, and recovery. Each sample was analyzed in seven replicates. A detailed description of the sample preparation process and the outcome of the validation study were presented in a previous study [13].

### 4.3. ED-XRF Analysis

Each sample was analyzed using an X-ray spectrometer Shimadzu, model EDX 700 (Kyoto, Japan). All the measurements were performed using a beam collimation of 10 mm, under air atmosphere. The following parameters were used: tube voltage of 15–50 kV; tube current of 100 µA; dead time of 30%. For spectral acquisition, bee pollen was placed into ED-XRF cells with polypropylene film (thickness of 5 µm). All measurements were repeated three times. Before every repetition, the samples were mixed inside the ED-XRF cells.

Samples collected from the same apiary (P1–P5) were analyzed in two forms: whole grains/granules and pollen crushed in a mortar into a powder. Two different ED-XRF methods were optimized to analyze bee pollen. Firstly, samples were analyzed using the quantitative Fundamental Parameter (FP) method with P5 as a standard sample. Further analysis was based on the qual–quantitative FP method.

The obtained results demonstrated that the quantitative ED-XRF method was the best suitable and proper approach for the analysis of whole pollen grains (based on a standard sample). Therefore, it was further used to analyze commercially available bee pollen samples.

### 4.4. ATR–FTIR Spectroscopy

The ATR–FTIR analysis was performed with a Shimadzu 8400 s spectrometer (Shimadzu Corp., Kyoto, Japan) applying an ATR accessory (Pike Technologies, Madison, WI, USA) with a germanium crystal. The bee pollen samples in the form of grains were put onto the crystal of the ATR and pressed with constant pressure. The selected wavenumber’s range was 750–4000 cm^−1^. Twenty scans were measured for each spectrum at the resolution of 2 cm^−1^. Before every measurement, a new background spectrum (a spectrum of air) was collected. All spectra were corrected on water and CO_2_, then saved in *.dx format in the instrument software IRSolution, and further exported to MATLAB.

### 4.5. Data Analysis

All the obtained ED-XRF data were collected in an Excel document. Mean concentrations values were calculated from three replicates. Precision values expressed as standard deviation (SD) and relative standard deviation (%RSD) were determined. The difference between the results obtained from various methods (ED-XRF and ICP-MS/ICP-OES) ware presented as percentage errors calculated as follows:percent error = [ED-XRF mean concentration value − ICP mean concentration value]/ICP mean concentration value × 100%(1)

Data were further exported to MATLAB and PLSToolbox. Data were preprocessed using log10 and autoscale transformation, and then hierarchical clustering analysis (CA) and Principal Component Analysis (PCA) were performed.

Data obtained from IR analysis were exported to MATLAB. All spectra in the range of 1800–900 cm^−1^ were preprocessed using a standard normal variable (SNV), second derivative, and autoscale. Further, CA and PCA analyses were conducted. Samples P1–P5 were also analyzed using two-trace two-dimensional (2T2D) correlation spectroscopy. Synchronous and asynchronous maps (with P1 as the reference sample) were generated.

## 5. Conclusions

The use of FTIR and ED-XRF spectroscopy combined with advanced chemometric tools was explored for the inexpensive and rapid analysis of bee pollen samples. The novel ED-XRF method was optimized to obtain a reliable inorganic samples characterization. The FTIR method and sophisticated 2T2D spectroscopy allowed us to obtain the complete picture of bee pollen organic components. The results confirmed that the composition of bee pollen might vary according to its geographic origin, the plant species from which the pollen was collected, or the harvest date. Our work indicates that bee pollen composition should be closely monitored for the presence and the content of individual compounds. Since the proposed approach demonstrates great potential, it should be employed in future studies of a larger group of bee pollen samples.

## Figures and Tables

**Figure 1 molecules-26-06024-f001:**
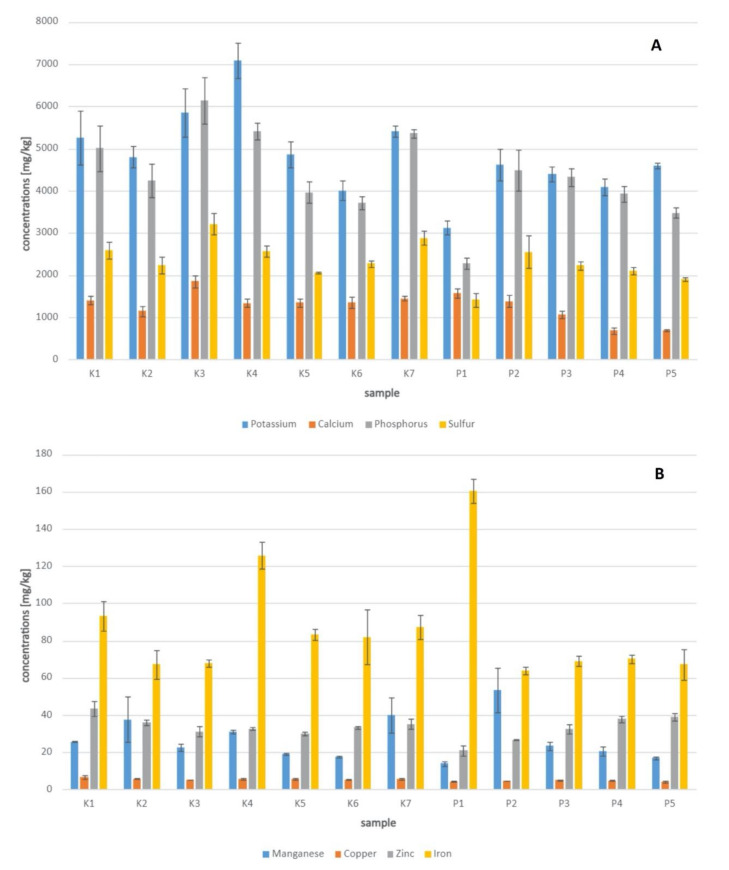
Concentrations of different elements (1.**A**.: K, Ca, P, S; 1.**B**.: Mn, Cu, Zn, Fe) measured with the ED-XRF quantitative method (using a standard sample) for bee pollen samples K1–K7 and P1–P5.

**Figure 2 molecules-26-06024-f002:**
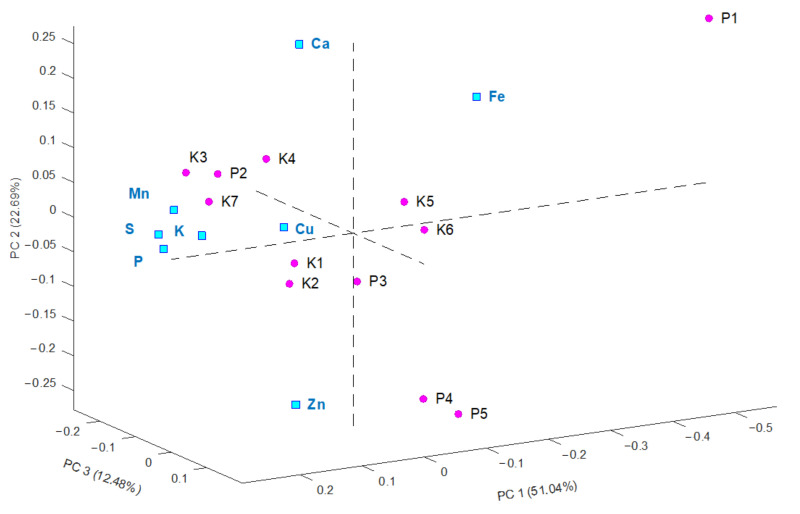
Biplot graph presenting the data from the ED-XRF analysis for bee pollen samples K1–K7 and P1–P5 (pink) and analyzed elements (blue).

**Figure 3 molecules-26-06024-f003:**
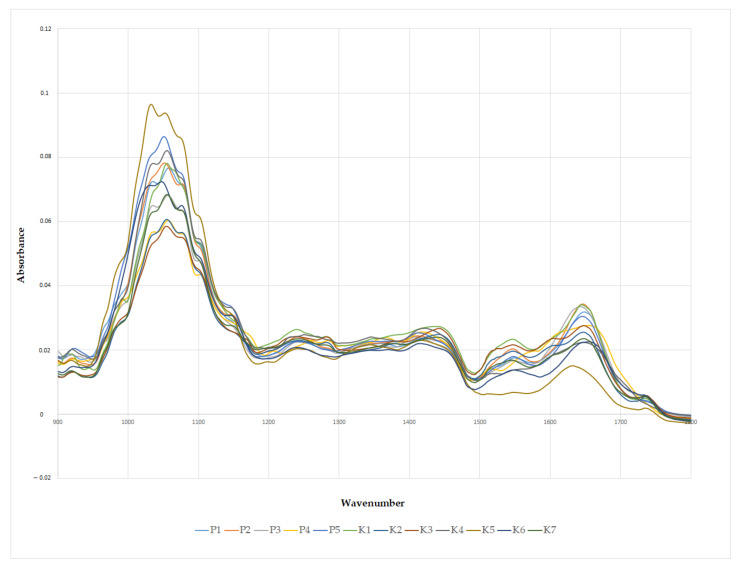
ATR–FTIR spectra of bee pollen samples (K1–K7; P1–P5) in the range of 1800–900cm^−1^.

**Figure 4 molecules-26-06024-f004:**
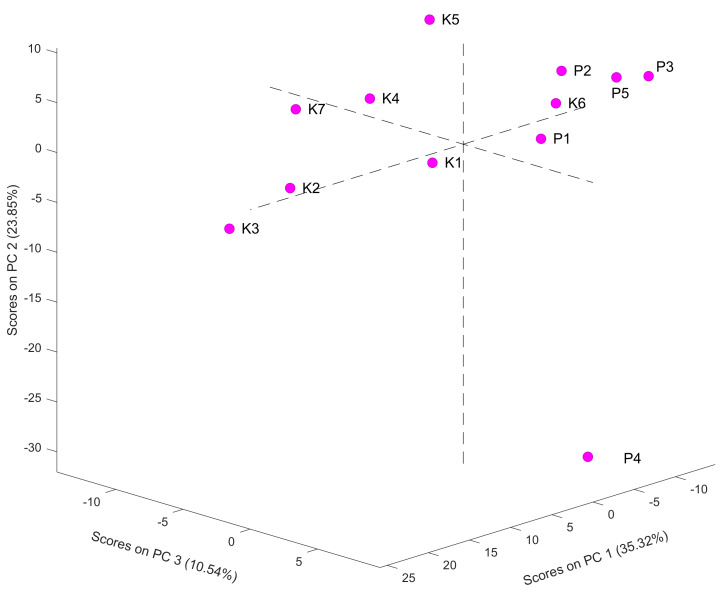
PCA analysis performed on the data from the ATR–FTIR analysis of bee pollen samples K1–K7 and P1–P5.

**Figure 5 molecules-26-06024-f005:**
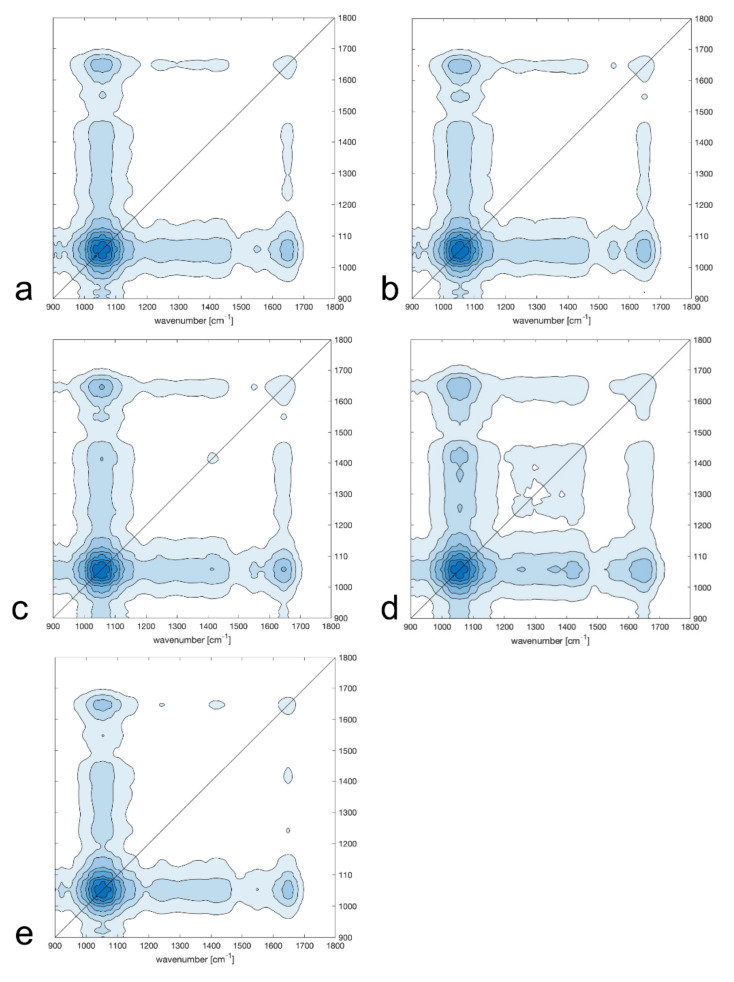
Synchronous maps for bee pollen samples collected on different dates: (**a**) 12 August 2018 (P1); (**b**) 2 June 2019 (P2); (**c**) 14 June 2019 (P3); (**d**) 29 June 2019 (P4), and (**e**) 20 July 2019 (P5).

**Figure 6 molecules-26-06024-f006:**
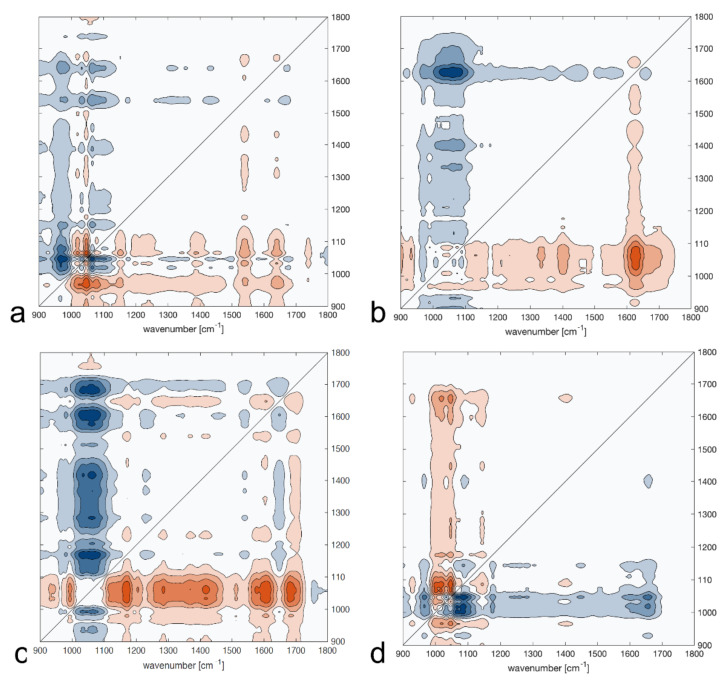
Asynchronous maps for bee pollen samples collected on different dates: (**a**) 2 June 2019 (P2); (**b**) 14 June 2019 (P3); (**c**) 29 June 2019 (P4), and (**d**) 20 July 2019 (P5) generated using sample P1 (from 12 August 2018) as a reference. The peaks with the positive sign are orange, and the negative peaks are blue.

**Table 1 molecules-26-06024-t001:** Comparison of the mineral content results obtained using different analytical methods (ED-XRF qualitative–quantitative, ED-XRF with standard sample, and ICP-MS/ICP-OES) for bee pollen samples collected on 2 June 2019 (P2) and 20 July 2019 (P5).

Mineral	Method	Sample (Date)	Mean + SD [mg/kg]	Method	Mean + SD [mg/kg]	Method	Mean [mg/kg]
Manganese	ED-XRF Qual-Quantitative analysis without standard sample	P2 (2 June 2019)	37.0 ± 7.1	ED-XRF Fundamental Parameter method with standard sample	53.3 ± 11.8	ICP-MS	62.0
Copper			8.1 ± 0.2		4.6 ± 0.2		5.7
Zinc			26.8 ± 4.9		26.6 ± 0.2		31.0
Potassium			4115.1 ± 263.9		4611.6 ± 379.6	ICP-OES	4600.0
Calcium			1131.4 ± 133.3		1376.5 ± 143.8		1500.0
Iron			48.0 ± 5.4		63.9 ± 2.2		49.0
Phosphorus			3085.9 ± 69.7		4482.5 ± 490.5		4600.0
Sulfur			2397.8 ± 224.4		2547.7 ± 385.7		2700.0
Chlorine			375.1 ± 72.7		-		-
Manganese	ED-XRF Qual-Quantitative analysis without standard sample	P5 (20 July 2019)	15.7 ± 2.1	ED-XRF Fundamental Parameter method with standard sample	17.0 ± 0.7	ICP-MS	16.0
Copper			8.6 ± 0.6		4.23 ± 0.4		4.4
Zinc			48.5 ± 0.4		39.0 ± 1.9		41.0
Potassium			4146.3 ± 27.3		4600.7 ± 71.6	ICP-OES	4400.0
Calcium			608.3 ± 14.1		697.9 ± 24.1		680.0
Iron			63.2 ± 4.8		67.2 ± 8.2		68.0
Phosphorus			2551.1 ± 80.1		3475.5 ± 124.4		3500.0
Sulfur			1915.2 ± 16.6		1907.5 ± 44.3		1900.0
Bromine			19.1 ± 2.7		-		-
Chlorine			615.1 ± 19.4		-		-

**Table 2 molecules-26-06024-t002:** Meaningful peaks in FTIR spectra of pollen samples; wavenumber (peak position) and corresponding vibrations and biochemical assignment.

Wavenumber (cm^−1^)	Vibrations and Biochemical Assignment
~1735	C=O stretching mode from lipids and hemicellulose
~1652	C=O stretching mode from amide I of proteins
~1546	N–H deformation mode and C–N stretching mode from amide II (shoulder)
~1514	C=C stretching mode from phenolic acids
~1420–1380	C–H deformation mode from lipids and cellulose
~1300–1200	N–H deformation mode and C–N stretching mode from amide III of proteins
~1180–950	C–O and C–C stretching mode from sugar and proteins

## Data Availability

The data presented in this study are contained within the article and Appendix A.

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
