# Peer review of "Rapid and Accurate Approach for Honeybee Pollen Analysis Using ED-XRF and FTIR Spectroscopy"

_molecules, 2021, doi:10.3390/molecules26196024_

Round 1
Reviewer 1 Report
This manuscript describes a new methodology that allows a comprehensive comparison of the components of multiple honeybee pollens. Two methods, ED-XRF and ATR-FTIR, are used for comprehensive analysis of the components, each of which is non-destructive and precise. By utilizing these methodologies, it is shown that it is possible to make an overall comparison of the components of inorganic and organic compounds with respect to unpurified honeybee pollen samples. The reviewer considers the content of this paper to be informative to the readers and therefore this paper is valuable to be published in this journal after considering the following points. 1. The letters and numbers in all figures are too small, so it is desirable to make them larger. 2. Regarding Figs. 7 and 8, it seems easy to understand if there is an instruction of P1-P5 in the figure. 3. It would be better to explain what red and blue colors mean in Fig 8. Minor points L330 “approach to monitoring” => “approach to monitor”? L344 “20 0C” => “0” should be superscript.Author Response
Dear Reviewer,
Hereby, we are sending a copy of an original research article revised according to the received comments. We thank the Reviewer for all the valuable remarks. Below, we would like to outline our responses to the reviewer’s comments.
Kindly note that all amendments in the manuscript are marked using the “Track changes” option in Microsoft Word.
Reviewer’s comments:
- The letters and numbers in all figures are too small, so it is desirable to make them larger.
Our response:
The quality of the figures was improved.
Reviewer’s comment:
- Regarding Figs. 7 and 8, it seems easy to understand if there is an instruction of P1-P5 in the figure.
- It would be better to explain what red and blue colors mean in Fig 8.
Our response:
An instruction of P1-P5 and the proper explanation were added.
Reviewer’s comment:
Minor points L330 “approach to monitoring” => “approach to monitor”? L344 “20 0C” => “0” should be superscript.
Our response:
The manuscript was carefully checked for some minor mistakes.
Once more, we appreciate your careful evaluation of our work and hope that this revision meets with your approval.
Apart from the abovementioned, several minor corrections were made in the manuscript:
- The abstract, introduction, and discussion were improved, and several references were added.
- Table 2, Figures 2 and 5 were moved to Supplementary Materials.
- Standard deviations were added in Figure 1.
- Figures presented PCA were improved, and additional table with detailed information about PCA was added to the Supplementary Materials.
Thank you again for your interest in our work. We await your review of our revised manuscript.
Yours sincerely,
Agata Swiatly-Blaszkiewicz
On behalf of the authors

Reviewer 2 Report
The manuscript entitled " Rapid and accurate approach for honeybee pollen analysis using ED-XRF and FTIR spectroscopy" discusses ED-XRF and ATR-FTIR methods to monitor the variability of the inorganic and organic content of the bee pollen.
A single hit to Scifinder Scholar with a search topic “Analysis of bee pollen” resulted in 19 references, although many more reports can be found with altering keywords. The ATR-FTIR spectroscopic analysis of bee pollen has recently been published [ACS Omega (2021), 6(7), 4878-4887]. This publication is a bit more informative compared to the submitted manuscript, although it is understandable that depending on biogeographical origin and type of flowers, the composition of the pollen may vary to some extent. I appreciate the experimental work in the paper, but I do not feel that this research throws new light that deserves publication in a journal devoted to new developments in fundamental and technological aspects of chemical sciences. On the other hand, the ICP-MS, and ICP-OES analyses of the pollen have been reported from the same research group recently [Molecules (2021), 26(9), 2415]. The ED-XRF analysis does not sound meaningful in the characterization of bee pollen, a complex mixture of several vitamins, minerals, carbohydrates, lipids, proteins, and many other components.
Therefore, considering side by side with the already published articles, the current findings are incremental and do not advance the field.
Author Response
Dear Reviewer,
Hereby, we are sending a copy of an original research article revised according to the received comments. We thank the Reviewer for all the valuable remarks. Below, we would like to outline our responses to the reviewer’s comments.
Kindly note that all amendments in the manuscript are marked using the “Track changes” option in Microsoft Word.
Reviewer’s comment:
A single hit to Scifinder Scholar with a search topic “Analysis of bee pollen” resulted in 19 references, although many more reports can be found with altering keywords. The ATR-FTIR spectroscopic analysis of bee pollen has recently been published [ACS Omega (2021), 6(7), 4878-4887]. This publication is a bit more informative compared to the submitted manuscript, although it is understandable that depending on biogeographical origin and type of flowers, the composition of the pollen may vary to some extent. I appreciate the experimental work in the paper, but I do not feel that this research throws new light that deserves publication in a journal devoted to new developments in fundamental and technological aspects of chemical sciences. On the other hand, the ICP-MS, and ICP-OES analyses of the pollen have been reported from the same research group recently [Molecules (2021), 26(9), 2415]. The ED-XRF analysis does not sound meaningful in the characterization of bee pollen, a complex mixture of several vitamins, minerals, carbohydrates, lipids, proteins, and many other components.
Our response:
We thank the Reviewer for this comment. The presented methodologies allowed to obtain “the whole picture” of the bee pollen components in a short time. The obtained results suggested that the proposed analytical strategy is proper for monitoring the quality of a large group of samples. The manuscript introduced a novel technique – ED-XRF for the bee pollen analysis and presented sophisticated correlation spectroscopy to characterize IR spectra better.
We agree that the recently published paper [ACS Omega (2021), 6(7), 4878-4887] is very informative and valuable. Therefore, we decided to mention it in our manuscript and discuss the results. The following sentences were added:
- Introduction
Lines: 89-100
In the literature, other analytical methods can be applied to the characterization of bee pollen composition [16–20]. One of the most frequently used is high-performance liquid chromatography (HPLC). It was proved to be an effective and reliable method to determine phenolic compounds [16] and amino acids [20] in bee pollen samples. Therefore, HPLC can be used as an authentication technique for this product. However, in this study, the main assumptions were the short analysis time and non-destructive character. Since FTIR enables the whole molecular profile screening, it seems to be the proper strategy for this research. Moreover, bearing in mind that macro-and micro-nutrients are essential for the customers, the quality of the bee pollen should also be examined regarding inorganic components. Therefore, ED-XRF and FTIR spectroscopy can be applied as complementary methods to monitor bee pollen variability.
- Zeghoud, S. et al. ACS Omega 2021
- Mazurek, S. et al. Antioxidants (Basel, Switzerland) 2021
- Paradowska, K. et al. J. Apic. Res. 2017
- Waś, E. et al. J. Apic. Sci. 2017
- Paramás, A.M.G. et al. Food Chem. 2006
- Discussion
Lines: 330-333
Recently, the novel approach for bee pollen analysis combing HPLC and IR was described [16]. Obtained IR-based metabolic profiles were compared with results derived from HPLC technique and subjected to multivariate analysis.
Following the reviewer's comments, we also decided to justify the use of ED-XRF instead of ICP-MS or ICP-OES:
- Introduction
Lines: 72-77
Recently our group has published a paper describing the use of ICP-MS and ICP-OES to analyze a large group of bee pollen samples, including propolis, bee pollen, and royal jelly [13]. However, comparing with ICP methods, ED-XRF is much faster and does not require tedious sample preparation. The use of ED-XRF enables the analysis of the most essential inorganic components. Therefore, it seems to be a powerful screening tool.
- Matuszewska, E. et al Molecules 2021
We hope that we convinced the Reviewer by our response that this manuscript is worth publication.
Apart from the abovementioned, several minor corrections were made in the manuscript:
- Table 2, Figures 2 and 5 were moved to Supplementary Materials.
- Standard deviations were added in Figure 1.
- Figures presented PCA were improved, and an additional table with detailed information about PCA was added to the Supplementary Materials.
- The manuscript was carefully checked for some minor mistakes and the significant digit of data.
Once more, we appreciate your careful evaluation of our work and hope that this revision meets with your approval.
Yours sincerely,
Agata Swiatly-Blaszkiewicz
On behalf of the authors

Reviewer 3 Report
The article discusses the qualitative analysis of the content of elements in various types of honey with the use of ED-XRF and FTIR spectroscopy.
After reading the manuscript, I would like to kindly ask you for more details or to correct some elements of the manuscript.
- The main attention is related to the concurrently applied cluster analysis and PCA analysis. The results of both analyzes are very similar to each other, so the results of one of these analyzes should be presented. Going further, the authors do not present the value of the loadings for the examined elements (Figure 3). Their presentation will make it possible to link the types of honey with the content of elements. Therefore, the results of the cluster analysis should be removed from the work and more attention should be paid to explaining the differences in the composition of elements between the types of honey.
- Figure 6: The components used explain less than 60% of the variability. What percentage of variability was explained by PC3 and PC4? Additional components should be included in the paper in order to increase the percentage of explained variability.
- Figure 7: What does the explanation “(a-e)” mean?
- Figure 8: What does the explanation “(a-d)” mean?
- Reference 34 in the bibliography should be corrected.
Author Response
Dear Reviewer,
Hereby, we are sending a copy of an original research article revised according to the received comments. We thank the Reviewer for all the valuable remarks. Below, we would like to outline our responses to the reviewer’s comments.
Kindly note that all amendments in the manuscript are marked using the “Track changes” option in Microsoft Word.
Reviewer’s comment:
The main attention is related to the concurrently applied cluster analysis and PCA analysis. The results of both analyzes are very similar to each other, so the results of one of these analyzes should be presented. Going further, the authors do not present the value of the loadings for the examined elements (Figure 3). Their presentation will make it possible to link the types of honey with the content of elements. Therefore, the results of the cluster analysis should be removed from the work and more attention should be paid to explaining the differences in the composition of elements between the types of honey.
Our response:
We agree with the Reviewer that the value of the loadings for the examined elements should be presented. Therefore, we prepared a biplot graph including samples and analyzed elements. Moreover, cluster analyses for ED-XRF and FTIR were moved to Supplementary Materials. Unfortunately, we do not exactly know the specific type of the analyzed pollens. The presented data and biplot suggest the diversity of the bee pollen composition regarding date and location of harvest, which is discussed in the manuscript.
Figure 3. Biplot graph presented data from ED-XRF analysis for bee pollen samples K1-K7 and P1-P5.
Figure 5. PCA analysis performed on the data from ATR-FTIR analysis for bee pollen samples K1-K7 and P1-P5.
Reviewer’s comment:
Figure 6: The components used explain less than 60% of the variability. What percentage of variability was explained by PC3 and PC4? Additional components should be included in the paper in order to increase the percentage of explained variability.
Our response:
Since the percentage of variability explained by PC1-PC4 are very important, PC3 was added to Figure 6, and the additional table was added to the Supplementary Materials:
Table S7. Detailed information about PCA analysis performed on the data from ED-XRF and ATR-FTIR (% Variance of the particular PC, % Variance cumulative, root mean squared error of calibration – RMSEC, and root mean squared error of cross-validation – RMSECV).
|
ED-XRF |
ATR-FTIR |
||||||
|
% Variance this PC |
% Variance Cumulative |
RMSEC |
RMSECV |
% Variance this PC |
% Variance Cumulative |
RMSEC |
RMSECV |
PC1 |
51.04 |
51.04 |
0.67 |
1.19 |
35.32 |
35.32 |
0.77 |
1.18 |
PC2 |
22.69 |
73.73 |
0.49 |
1.34 |
23.85 |
59.17 |
0.61 |
1.16 |
PC3 |
12.48 |
86.21 |
0.36 |
1.37 |
10.54 |
69.71 |
0.53 |
1.17 |
PC4 |
6.13 |
92.34 |
0.26 |
1.81 |
8.40 |
78.11 |
0.45 |
1.19 |
Reviewer’s comment:
Figure 7: What does the explanation “(a-e)” mean?
Figure 8: What does the explanation “(a-d)” mean?
Reference 34 in the bibliography should be corrected.
Our response:
The proposed changes were made in the manuscript: the figures explanations and references were corrected.
Apart from the abovementioned, several minor corrections were made in the manuscript:
- The abstract, introduction, and discussion were improved, and several references were added.
- Table 2, Figures 2 and 5 were moved to Supplementary Materials.
- Standard deviations were added in Figure 1.
- The manuscript was carefully checked for some minor mistakes.
Once more, we appreciate your careful evaluation of our work and hope that this revision meets with your approval.
Yours sincerely,
Agata Swiatly-Blaszkiewicz
On behalf of the authors

Reviewer 4 Report
In this study, two non-destructive spectroscopy methods, including ED-XRF and ATR-FTIR, were proposed as fast approaches to characterize bee pollen. Here are some suggestions for further improvement of the manuscript:
- The abstract can be improved. The background introduction can be shortened, and the key results of present study can be described in detail.
- The introduction section can be improved. Please describe clearly why the authors chose ED-XRF and ATR-FTIR methods together for the analysis of bee pollen. The other techniques such as chromatographic methods for the analysis of bee pollen can be introduced.
- The quality of figure 1 and 4 can be improved. Figure 6 can be moved to Supplementary Materials.
- Table 1 can be moved to Supplementary Materials, as the data have been shown in Figure 1. Table 3 can be deleted and describe the content in the main text.
- Please check the significant digit of the data, such as “10%” (Line 160) and “0.37%-25.62%” (Line 144); “750-4000 cm-1”(Line 386) and “1514cm-1, 1735cm-1”(Line 242).
- Please check carefully for some minor mistakes. Such as “were” (Line 141), “demonstrates” (150) and “are” (Line 157,193 and 224). “-20 0C” (Line 344)?
Author Response
Dear Reviewer,
Hereby, we are sending a copy of an original research article revised according to the received comments. We thank the Reviewer for all the valuable remarks. Below, we would like to outline our responses to the reviewer’s comments.
Kindly note that all amendments in the manuscript are marked using the “Track changes” option in Microsoft Word.
Reviewer’s comment:
The abstract can be improved. The background introduction can be shortened, and the key results of present study can be described in detail.
Our response:
As suggested by the Reviewer, the following sentences were added to the Abstract:
Lines: 26-29
Developed ED-XRF method allowed to obtain reliable quantification of 8 macro- and micro-nutrients, while organic components were characterized by IR spectroscopy. Principal component analysis, cluster analysis, and obtained synchronous and asynchronous maps allowed to study component changes occurring due to date and location of harvest.
Reviewer’s comment:
The introduction section can be improved. Please describe clearly why the authors chose ED-XRF and ATR-FTIR methods together for the analysis of bee pollen. The other techniques such as chromatographic methods for the analysis of bee pollen can be introduced.
Our response:
We thank the Reviewer for this comment. The introduction was improved, and the following sentences were added:
Lines: 72-77
Recently our group has published a paper describing the use of ICP-MS and ICP-OES to analyze a large group of bee pollen samples, including propolis, bee pollen, and royal jelly [13]. However, comparing with ICP methods, ED-XRF is much faster and does not require tedious sample preparation. The use of ED-XRF enables the analysis of the essential inorganic components. Therefore, it seems to be a significant screening tool.
Lines: 89-100
In the literature, other analytical methods can be applied to the characterization of bee pollen composition [16–20]. One of the most frequently used is high-performance liquid chromatography (HPLC). It was proved to be an effective and reliable method to determine phenolic compounds [16] and amino acids [20] in bee pollen samples. Therefore, HPLC can be used as an authentication technique for this product. However, in this study, the main assumptions were short time of the analysis and non-destructive character. Since FTIR enables screening of the whole molecular profile, it seems to be the proper strategy for this research. Moreover, bearing in mind that macro- and micro-nutrients are very important for the customers, the quality of the bee pollen should also be examined regarding inorganic components. Therefore, ED-XRF and FTIR spectroscopy can be applied as complementary methods to monitor bee pollen variability.
- Matuszewska, E. et al Molecules 2021
- Zeghoud, S. et al. ACS Omega 2021
- Mazurek, S. et al. Antioxidants (Basel, Switzerland) 2021
- Paradowska, K. et al. J. Apic. Res. 2017
- Waś, E. et al. J. Apic. Sci. 2017
- Paramás, A.M.G. et al. Food Chem. 2006
Reviewer’s comment:
The quality of Figures 1 and 4 can be improved. Figure 6 can be moved to Supplementary Materials.
Our response:
The quality of the figures was improved. According to the suggestions of Reviewer 3., Figures 2 and 5 were moved to Supplementary Materials. The results of cluster analysis and PCA are very similar, and due to Reviewer 3., we should pay more attention to PCA.
Reviewer’s comment:
Table 1 can be moved to Supplementary Materials, as the data have been shown in Figure 1. Table 3 can be deleted and describe the content in the main text.
Our response:
We agree with the Reviewer that the data in Table 2 and Figure 1 are the same. Therefore, Table 2 was moved to the Supplementary Materials (Supplementary Materials Table S6).
Table 3 was not deleted from the manuscript. According to other papers concerning IR analysis, it is clearer to present bands, their corresponding vibrations and biochemical assignment in the table. However, the most important peaks are described in the main text.
Reviewer’s comment:
Please check the significant digit of the data, such as “10%” (Line 160) and “0.37%-25.62%” (Line 144); “750-4000 cm-1”(Line 386) and “1514cm-1, 1735cm-1”(Line 242).
Please check carefully for some minor mistakes. Such as “were” (Line 141), “demonstrates” (150) and “are” (Line 157,193 and 224). “-20 0C” (Line 344)?
Our response:
The manuscript was carefully checked for some minor mistakes and the significant digit of data.
Apart from the abovementioned, several minor corrections were made in the manuscript:
- The discussion was improved, and several references were added.
- Standard deviations were added in Figure 1.
- Figures presented PCA were improved, and additional table with detailed information about PCA was added to the Supplementary Materials.
Once more, we appreciate your careful evaluation of our work and hope that this revision meets with your approval.
Yours sincerely,
Agata Swiatly-Blaszkiewicz
On behalf of the authors
Reviewer 5 Report
Tables 1 and 2 must be improved; please include standard deviations and a footer. In addition, the number of digits to the right of the decimal point in Tables is excessive.
Figure 1 must be edited, and standard deviations must be included.
Figure 4 must be improved. Is it absorbance in Y axe?
Please describe Principal component analysis (PCA) in 2.1.4 subheading.
Author Response
Dear Reviewer,
Hereby, we are sending a copy of an original research article revised according to the received comments. We thank the Reviewer for all the valuable remarks. Below, we would like to outline our responses to the reviewer’s comments.
Kindly note that all amendments in the manuscript are marked using the “Track changes” option in Microsoft Word.
Reviewer’s comment:
Tables 1 and 2 must be improved; please include standard deviations and a footer. In addition, the number of digits to the right of the decimal point in Tables is excessive.
Our response:
As suggested by the Reviewer, Tables 1 and 2 were improved. Additionally, Table 2 was moved to Supplementary Materials as the data have been shown in Figure 1.
Reviewer’s comment:
Figure 1 must be edited, and standard deviations must be included.
Figure 4 must be improved. Is it absorbance in Y axe?
Our response:
Figures 1 and 4 were improved.
Reviewer’s comment:
Please describe Principal component analysis (PCA) in 2.1.4 subheading.
Our response:
The proposed changes were made in the manuscript.
Apart from the abovementioned, several minor corrections were made in the manuscript:
- The abstract, introduction, and discussion were improved, and several references were added.
- Table 2, Figures 2 and 5 were moved to Supplementary Materials.
- Figures presented PCA were improved, and an additional table with detailed information about PCA was added to the Supplementary Materials.
- The manuscript was carefully checked for some minor mistakes.
Once more, we appreciate your careful evaluation of our work and hope that this revision meets with your approval.
Yours sincerely,
Agata Swiatly-Blaszkiewicz
On behalf of the authors
Round 2
Reviewer 3 Report
After the changes have been made, I suggest accepting the work for publication.
Reviewer 4 Report
The comments have been well addressed, and the manuscript has been carefully revised. It may be accepted for publication.